# Monolithically integrated, broadband, high-efficiency silicon nitride-on-silicon waveguide photodetectors in a visible-light integrated photonics platform

Yiding Lin [1] ✉, Zheng Yong[2], Xianshu Luo [3], Saeed Sharif Azadeh[1], Jared C. Mikkelsen[1], Ankita Sharma[1,2], Hong Chen[1], Jason C. C. Mak[2], Patrick Guo-Qiang Lo[3], Wesley D. Sacher[1] & Joyce K. S. Poon [1,2] ✉

Visible and near-infrared spectrum photonic integrated circuits are quickly becoming a key technology to address the scaling challenges in quantum information and biosensing. Thus far, integrated photonic platforms in this spectral range have lacked integrated photodetectors. Here, we report silicon nitride-on-silicon waveguide photodetectors that are monolithically integrated in a visible light photonic platform on silicon. Owing to a leaky-wave silicon nitride-on-silicon design, the devices achieved a high external quantum efficiency of >60% across a record wavelength span from $\lambda \sim 400$ nm to ~640 nm, an opto-electronic bandwidth up to 9 GHz, and an avalanche gain-bandwidth product up to $173 \pm 30$ GHz. As an example, a photodetector was integrated with a wavelength-tunable microring in a single chip for on-chip power monitoring.

Silicon (Si) photonics leverages microelectronics fabrication in foundries to mass manufacture dense and complex photonic integrated circuits (PICs). Extending Si photonics into the visible and near-infrared (NIR) spectrum ($\lambda \sim 400$–800 nm)[1–5] can bring PICs to an even wider range of emerging applications, including spectroscopy and flow cytometry[6,7], neurophotonics[8–10], quantum information processing[5,11,12], underwater communication[13], and scanning displays[14–16]. In Si photonics for the visible and NIR wavelength range, the optical waveguide is typically formed in a silicon nitride (SiN) or aluminum oxide ($Al_2O_3$) layer surrounded by a silicon dioxide ($SiO_2$) cladding on a Si or silicon-on-insulator (SOI) substate. Thus far, on 100, 200, or 300-mm diameter Si wafers, visible/NIR photonic platforms with SiN[3,5,17–21] or $Al_2O_3$[3,4] waveguides are usually passive, containing components such as gratings, power splitters, and interferometers, with waveguide phase-shifters tuned by the thermo-optic or strain-optic effect[5,22–24]. Post-fabrication processing and heterogeneous integration steps are used to incorporate liquid crystal phase-shifters[25], lasers[26,27], and photodetectors (PDs)[28,29] with SiN waveguides.

On-chip optical-to-electrical conversion is an essential functionality in a PIC platform. In visible and NIR Si photonics, the Si substrate serves as a natural candidate for light absorption compatible with monolithic integration, circumventing heterogeneous integration that increases the fabrication complexity. Si PDs are conventionally surface incident, but in PICs, waveguide PDs are preferred since they can be easily integrated with other in-plane circuit components. The main challenge in realizing a broadband, high-efficiency waveguide PD is the input waveguide-to-Si coupling, which can exhibit significant wavelength dependence across the visible and NIR spectral bands. One approach is to use the SOI device layer for photodetection and an overhead waveguide to achieve leaky-wave coupling. However, such designs are complicated by the presence of multiple Si slab modes that hybridize with the waveguide mode when their effective indices are

[1]Max Planck Institute of Microstructure Physics, Weinberg 2, 06120 Halle, Germany. [2]Department of Electrical and Computer Engineering, University of Toronto, 10 King's College Road, Toronto, Ontario M5S 3G4, Canada. [3]Advanced Micro Foundry Pte Ltd, 11 Science Park Road, Singapore Science Park II, 117685 Singapore, Singapore. ✉e-mail: yidinlin@mpi-halle.mpg.de; joyce.poon@mpi-halle.mpg.de

similar, resulting in resonance-like peaks in the absorption spectrum at wavelengths where the phase-matching condition is met[30–32]. Another approach is to end-fire couple the input waveguide to a Si absorption region as in[33], but this complicates the fabrication by requiring the input waveguide and Si to be co-planar, and the insertion loss is wavelength-dependent due to the spectral variation of the mode field diameter mismatch between SiN and Si.

Here, we report broadband, high-efficiency, SiN-on-Si waveguide PDs integrated within a foundry-fabricated visible spectrum PIC platform fabricated on 200-mm Si wafers. Preliminary results of the devices were reported in[34]. Figure 1a shows the PIC platform cross-section, which comprises two layers of SiN waveguides, a titanium nitride (TiN) heater, and metal layers on bulk Si. Components from this platform, such as the suspended thermo-optic phase-shifters, bi-layer fiber-to-chip couplers, and electro-thermally actuated micro-electro-mechanical system cantilevers, are reported elsewhere[23,35–37]. The SiN-on-Si PDs have a leaky-wave design where the optical power in the SiN input waveguide evanescently leaks into a mesa defined in bulk Si under the waveguide. The thick and wide Si mesa effectively supports a continuum of radiation modes, eliminating the coupling sensitivity to the phase-matching condition in SOI-based devices and facilitating broadband and efficient power transfer. The PD geometry is alignment tolerant. We achieved a record operating wavelength span of >230 nm, with an external quantum efficiency (EQE) > 60% from $\lambda \sim$ 400–640 nm for the transverse magnetic (TM) polarization. The PDs exhibited opto-electronic (OE) bandwidths up to 9 GHz and also operated as avalanche photodiodes (APDs) with a gain-bandwidth product up to $173 \pm 30$ GHz. Lastly, as a proof of concept, we demonstrate the integration of the PD with a tunable microring filter in a single chip for on-chip power monitoring.

## Results

The dashed box in Fig. 1a shows the cross-section of the PD design, which is comprised of a SiN waveguide passing atop a lateral PIN or PN junction in a Si mesa. Unless otherwise stated, all indicated dimensions are nominal values. For PIN junctions, the intrinsic region had a width of 2 μm, centered below the SiN waveguide. The gap ($W_{gap}$) between the SiN and Si layer was designed to be 150 nm. Two layers of metals and vias connected the device to bond pads. The vias and metals were placed sufficiently apart to ensure optical isolation from the SiN

waveguide. The Si mesa height was 2.85 μm to achieve a low propagation loss for the routing SiN waveguides (SiN1 layer in Fig. 1a). Figure 1b shows a scanning electron micrograph of the PD cross-section with the SiN waveguide atop the Si mesa in the inset, and Fig. 1c shows an optical micrograph of a device.

Figure 1d shows the top-view schematic of a PD device with a reference waveguide structure for optical power calibration during characterization. As designed, the routing SiN waveguides had a thickness of $t = 150$ nm and widths ($W_{gw}$) of 500 nm (for the single PD devices) and 380 nm (for the microring integration). Tapered edge couplers, starting with a SiN width of 5.2 μm that adiabatically narrowed to $W_{gw}$ over a length of 300 μm, were used for fiber-to-chip coupling. The routing waveguides adiabatically narrowed over a length of 100 μm to $W_{gn}$ (i.e., 150, 200, and 250 nm) at the Si mesa facet for light to penetrate into the Si. We have previously reported the propagation loss of the routing waveguides (-1.1–5.9 dB/cm from $\lambda = 430$–648 nm[2], and -1.8–7.1 dB/cm from $\lambda = 405$–640 nm[35]) and insertion loss of the edge couplers (7.5–11.3 dB/facet from $\lambda = 430$–648 nm[2], and 5.9–8.6 dB/facet from $\lambda = 405$–640 nm[35]). The corresponding measurements for this work are in the Supplementary Information (Fig. S2).

Figure 2a shows optical micrographs of the same example device under test at different wavelengths. The images have not been post-processed. In our design, the input facet of the Si mesa was at the middle of the chip (L/2, Fig. 1d), and the input power to the PD, $P_{in,PD}$, was thus taken as $(P_{in,ref} + P_{out,ref})/2$ using a reference waveguide with the same width $W_{gw}$, where the $P_{in,ref}$ and $P_{out,ref}$ are the measured optical powers (in dBm) into and out of the waveguide, respectively.

### PIN Photodiodes

First, we report the PIN PDs with $l = 50$ μm, and $W_{gn} = 200 \pm 20$ nm, $t = 120 \pm 10$ nm, and $W_{gap} = 190 \pm 10$ nm as measured from cross-sectional transmission electron microscope images, and TE polarized input unless otherwise stated (Fig. 2b–e). Figure 2b shows the measured current-voltage ($I$–$V$) characteristics of a device at $\lambda = 488$ nm. Measurements from >20 devices distributed uniformly on the wafer showed that the dark currents were, on average, $144 \pm 42$ pA and $266 \pm 65$ pA at reverse biases, $V_r$, of −5 and −15 V. No observable avalanche breakdown was found up to −20 V. At $\lambda = 488$ nm, a photocurrent of -1.35 μA was observed at a $P_{in,PD}$ of $-23.9 \pm 0.3$ dBm at −2 V

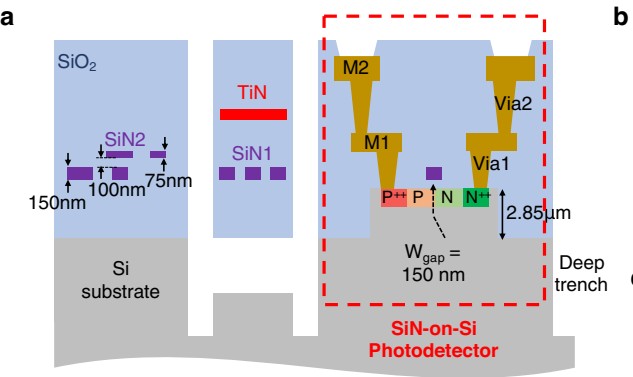

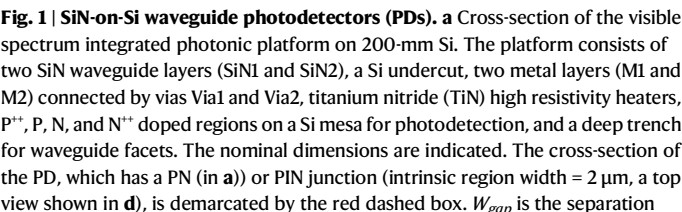

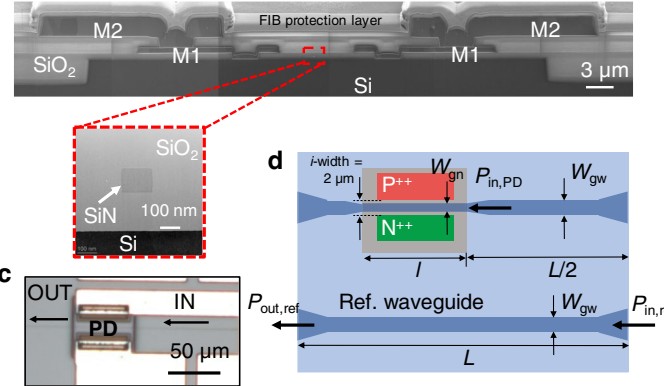

**Fig. 1 | SiN-on-Si waveguide photodetectors (PDs). a** Cross-section of the visible spectrum integrated photonic platform on 200-mm Si. The platform consists of two SiN waveguide layers (SiN1 and SiN2), a Si undercut, two metal layers (M1 and M2) connected by vias Via1 and Via2, titanium nitride (TiN) high resistivity heaters, P$^{++}$, P, N, and N$^{++}$ doped regions on a Si mesa for photodetection, and a deep trench for waveguide facets. The nominal dimensions are indicated. The cross-section of the PD, which has a PN (in **a**) or PIN junction (intrinsic region width = 2 μm, a top view shown in **d**), is demarcated by the red dashed box. $W_{gap}$ is the separation

between the SiN1 layer and Si in the PD. **b** A stitched scanning electron micrograph of the PD cross-section. A protective layer had been applied for the focused ion beam (FIB) milling to prepare the sample for imaging. Inset: Transmission electron micrograph showing the SiN waveguide above the Si. **c** An optical micrograph of a PD with an input SiN waveguide. **d** Top-view schematic of the PD test structure with a reference waveguide for power calibration. $P_{in,ref}$ and $P_{out,ref}$ are reference input and output optical powers; $W_{gw}$ is the width of the routing waveguides; and $P_{in,PD}$ is the input optical power into the PD.

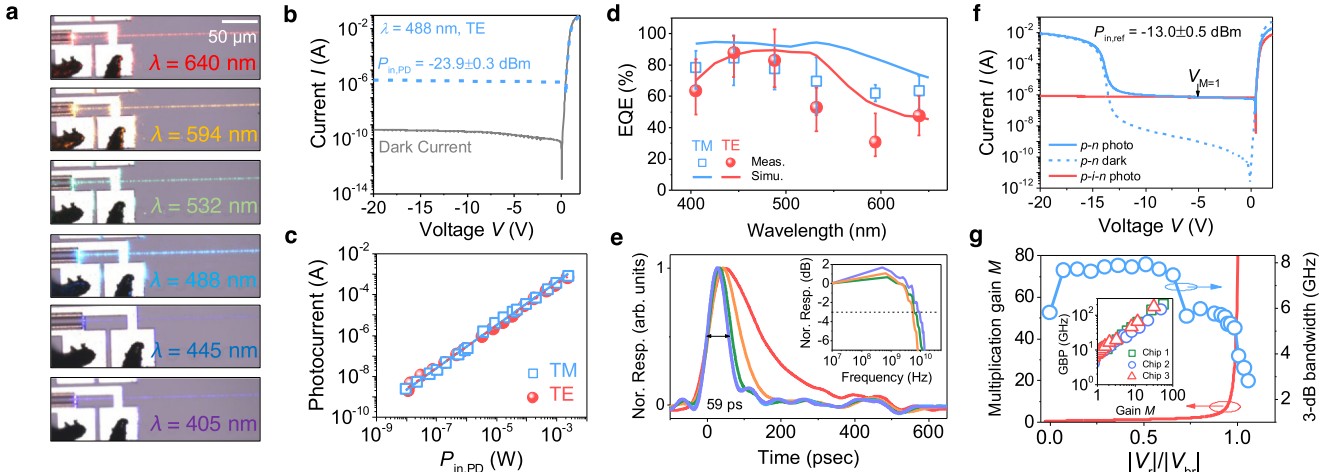

**Fig. 2 | Characterization of the SiN-on-Si PDs (device length: 50 μm). a** Optical micrographs of the same device under test at different wavelengths. (**b-e**) Characterization of PIN devices ($l$ = 50 μm, measured $W_{gn}$ = 200 ± 20 nm, $t$ = 120 ± 10 nm, and $W_{gap}$ = 190 ± 10 nm). **b** Current–voltage ($I$–$V$) characteristics without (dark current) and with $P_{in,PD}$ of −23.9 ± 0.3 dBm. **c** $I_{eph}$ as a function of $P_{in,PD}$ from - $10^{-9}$ to $10^{-3}$ W. **d** Measured EQE at −2 V agrees well with the simulation. Three devices from three chips far apart (>10 cm) on the wafer were measured. The error bars show the maximum and minimum quantities measured for the three devices, and the data points show the average. **e** Normalized impulse response of the device at different reverse biases. Inset: normalized frequency response extracted from fast Fourier transform. Legend: red, 0 V; yellow, −2 V; green, −5 V; blue, −10 V; and violet, -20 V. The data in **b**, **c** and **e** came from one of the devices in **d**. (**f-g**) Characterization of PN junction APDs. **f** $I$–$V$ characteristics without and with $P_{in,ref}$ = −13.0 ± 0.5 dB m at $\lambda$ = 405 nm, before coupling into the SiN waveguide. The data for the PIN device came from one of the devices in **d**. **g** The avalanche multiplication gain ($M$) and 3-dB OE bandwidth as a function of $|V_r|/|V_{br}|$. Inset shows the gain-bandwidth product (GBP) of devices from three dies on the wafer.

and was maintained throughout the $V_r$ range. The photocurrent included the absorption of the stray portion of $P_{in,ref}$ that was not coupled into the waveguide, which was eliminated by fiber displacement measurements (see Methods). We measured the effective photocurrent, $I_{eph}$, due to the light coupled into the waveguide without the stray light (as defined in Methods, Eq. (1)) over six orders of magnitude of $P_{in,PD}$ (-$10^{-9}$–$10^{-3}$ W) and found good linearity in the photocurrent response (Fig. 2c). The measurements of a PN device are included in the Supplementary Information (Fig. S3), and they also showed good $I_{eph}$–$P_{in,PD}$ linearity.

The EQE was calculated as $\mathcal{R}hc/(\lambda q)$, where $\mathcal{R}$ = $I_{eph}/P_{in,PD}$ is the responsivity and $h$, $c$, $q$ denote Planck's constant, the speed of light, and the elementary charge, respectively. Figure 2d shows the measured EQE at −2 V. The measured EQE qualitatively agrees with but is lower than the simulated results, which assume an ideal internal quantum efficiency and lossless SiN waveguides and transitions (see Supplementary Information). The EQE was >60% for a spectral span of -135 nm (400–535 nm) and >230 nm (400–640 nm) for the transverse electric (TE) and TM polarized mode, respectively. For $\lambda \gtrsim 550$ nm, although the mode approached cut off at $W_{gn}$ = 200 nm (Supplementary Information, Fig. S5c), the EQE benefits from the absorption of scattered light (Supplementary Information, Fig. S8). The measured EQE is limited by a number of factors, such as the internal quantum efficiency, mode mismatch loss between the input SiN waveguide and the SiN-on-Si region, $W_{gn}$ variations, device length, scattering and back-reflection at the Si mesa interface, and excitation of high order modes in the routing waveguide at short wavelengths (e.g., near 405 nm) (Supplementary Information). Nevertheless, Fig. S7 (Supplementary Information) shows a device length of $l$ = 50 μm is sufficient to saturate the EQE for $\lambda$ > 445 nm. We verified that longer devices (up to $l$ = 500 μm) showed an increase of <30% in $\mathcal{R}$ (Supplementary Information, Fig. S9). Increasing $l$ or narrowing $W_{gn}$ can increase the EQE for the TE polarization at $\lambda$ < 450 nm (Supplementary Information, Fig. S10). Simulations show that the EQE is insensitive to the Si width, Si thickness, and $W_{gap}$ near our experimental value of $W_{gap}$ = 190 nm (Supplementary Information, Figs. S11 and S12). $W_{gap} \in [110, 220]$ nm maximizes the EQE for both polarizations.

We extracted the PD OE bandwidth by inputting short pulses (full-width-at-half-maximum (FWHM) pulse width < 50 ps) at a wavelength of 405 nm into the device and measuring the output voltage in real-time (see Methods). Figure 2e displays the measured impulse response of a device at different $V_r$ (the line labels are in the caption). The FWHM decrease with increasing $|V_r|$ can be attributed to the reduced carrier transit time due to an electric field in the depletion region, as well as the reduced junction capacitance due to a wider depletion width. The frequency response was obtained by applying the Fourier transform on the impulse response, as shown in the inset of Fig. 2e. The OE 3-dB bandwidth was 4.4 ± 1.1 and 8.6 ± 1.0 GHz at −2 and −20 V, respectively. The measured PIN junction capacitance and contact resistance (Supplementary Information, Figs. S13, 14) suggested that the measured bandwidth was limited by the carrier transit time at low bias ($|V_r|$ < 10 V), and the laser pulse width and instrumentation at high bias ($|V_r|$ > 10 V) (Supplementary Information, Fig. S16).

## PN avalanche photodiodes

The PN junction PDs could also function as APDs. Figure 2f shows the $I$–$V$ characteristics of such a device ($l$ = 50 μm and measured $W_{gn}$ = 200 ± 20 nm, $t$ = 120 ± 10 nm, $W_{gap}$ = 190 ± 10 nm). Avalanche multiplication was significant beyond $V_r$ - −14 V. An $I_{ph}$−V characteristic of an identical PIN device with same $P_{in,ref}$ was used as a reference to determine the unity gain point, since $I_{ph}$ for the PIN device has been verified without avalanche gain up to −20 V (Fig. 2b). The unity gain voltage ($V_{M=1}$) was determined as the intersection point of the $I_{ph}$−V curves between the PN and PIN devices. Measurements from three chips across the wafer showed $V_{M=1}$ = − 4.5 ± 0.9 V for $l$ = 50 μm devices. We then found an avalanche multiplication gain, $M$, of 46 ± 14 and a corresponding gain-bandwidth product (GBP) of 173 ± 30 GHz at the avalanche breakdown voltage, $V_{br}$ (Fig. 2g). Details on the determination of $M$, GBP, and $V_{br}$ are in the "Methods" section and Supplementary Information, as well as the performance of APDs with $l$ = 100 μm.

## A tunable microring integrated with a PD

Finally, as a proof-of-concept PIC demonstration, we integrated a PIN PD at the tap of the through port of a tunable SiN racetrack microring

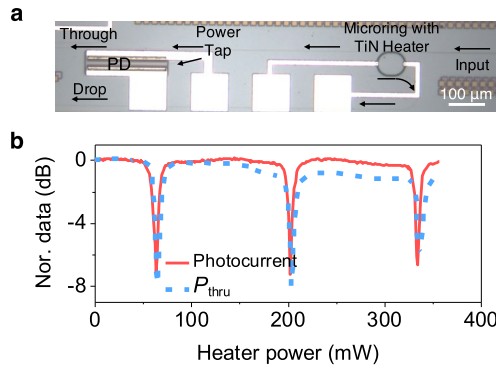

**Fig. 3 | On-chip characterization of a racetrack microring filter using an integrated PD. a** An optical micrograph of the PIC consisting of a thermally tunable microring with a PD connected to the power tap at the through output port. **b** Normalized photocurrent and through port transmission, $P_{thru}$, as a function of the TiN heater power.

filter. Figure 3a shows an optical micrograph of the PIC. The TiN heater above the ring was used for thermal tuning, and the microring did not use the suspended heater structure in ref. 23 which required clearance around the waveguide. We simultaneously measured the output power at the through port and the photocurrent while tuning the ring for an input wavelength of 514 nm. Details of the measurement are described in the Methods section. Figure 3b depicts a good match between the normalized through port transmission, $P_{thru}$, and photocurrent, demonstrating the feasibility of the PD for on-chip power monitoring. The slight increase in the insertion loss observed at the through port with the increase of the heater power was likely due to a drift of the output coupling, since the insertion loss increase was not observed by the on-chip PD. To our knowledge, this is the first visible spectrum photonic circuit with a monolithically integrated photodetector.

## Discussion

Our monolithically integrated SiN-on-Si waveguide PDs are unique and represent a new state of the art. Compared to the Si PD end-fired coupled to a SiN waveguide in ref. 33, which was characterized only at 685 nm, our SiN-on-Si PDs exhibited a higher EQE across a significantly wider wavelength span while maintaining a similar GBP ($173 \pm 30$ GHz in this work vs. $234 \pm 25$ GHz in ref. 33) and wafer-level performance uniformity (up to $\pm 30\%$ variation on dark current, EQE, $M$, OE bandwidth and GBP). The PDs reported here also have superior EQE and OE bandwidth compared to the heterogeneously integrated PDs in refs. 29 and 28. Si metal-semiconductor-metal (MSM) PDs integrated on SiN waveguides as in refs. 38 and 39 have ~kHz OE bandwidths and are significantly slower than the PN and PIN devices demonstrated here. A comparison table is available in the Supplementary Information.

In summary, we demonstrated PDs monolithically integrated into a visible spectrum Si PIC platform. The SiN-on-Si PDs achieved broadband high-efficiency photon detection across the entire visible spectrum. Due to a leaky-wave design of a tapered SiN waveguide over a Si mesa, an EQE > 60% was achieved over a wide wavelength span from 400 to 640 nm (for TM polarized light). The PDs had a 3-dB OE bandwidth up to 9 GHz and could also be used as APDs. Compared to other waveguide-integrated visible and NIR PDs, the device has the widest operating wavelength range while exhibiting similar dark current and APD gain-bandwidth products. The monolithically integrated SiN-on-Si PDs open the path toward sophisticated NIR and visible PICs. Future work includes the integration of PD arrays with an additional well doping opposite to the substrate doping type to avoid current leakage among devices and

the investigation of the device operation as single-photon APDs (SPADs).

## Methods

### Fabrication

The integrated photonic platform was fabricated on 200-mm Si substrates at Advanced Micro Foundry. The fabrication begins with forming the PDs. First, doped regions were formed by ion implantation followed by rapid thermal annealing, and then mesas were etched. Next, the SiO$_2$ bottom cladding was deposited. Two SiN waveguide layers were then formed by plasma-enhanced chemical vapor deposition, patterned by ArF deep ultraviolet lithography and reactive ion etching. Chemical mechanical polishing was used for layer planarization. The metal and heater layers were then defined followed by deep trench and undercut etching for suspended structures and edge coupler facets.

Ellipsometry measurements show the SiN and cladding SiO$_2$ had refractive indices of ~1.82 and ~1.46, respectively, at $\lambda = 488$ nm.

### DC characterization setup

For the responsivity and EQE measurement, the fabricated photodiodes were characterized using a continuous-wave multi-wavelength source (Coherent OBIS Galaxy). Cleaved Nufern S405-XP single-mode fiber was used to edge-couple the light into and out of the chip. The input and output fibers were mounted on 5-axis piezo-controlled micro-manipulators for precise alignment. A polarization controller (Thorlabs CPC900) was inserted into the input fiber path for the polarization control. Optical power was measured using a free-space wand-style power detector (Newport 918D-ST-SL, for $P_{in,ref}$ measurement) and a fiber-optic detector (Newport 818-SL-L, for $P_{out,ref}$ measurement), and was then read out by a power meter (Newport 2936-R). $I$–$V$ characteristics were obtained using a Keysight B2912A Precision Source/Measure Unit (SMU) (source voltage and measure current) via two DC electrical probes (MPI MP40 micropositioner, tungsten probe tip) placed on device anode and cathode contact pads. The voltage sweep ranged from forward (+2 V) to reverse (−20 V) bias in steps of 0.02 V. We verified that the photocurrent from ambient light was negligible during the dark current sweeps. Each photocurrent sweep was carried out at the respective maximum photocurrents by performing careful fiber alignment with the input SiN waveguides.

The photocurrent also included the absorption of stray light from $P_{in,ref}$ that was not coupled into the waveguide. To estimate this contribution, the fiber was horizontally displaced ~3 μm from the optimal fiber-waveguide alignment to eliminate the waveguide-coupled light absorption (mode field diameter ~3 μm for Nufern S405-XP). The corresponding photocurrent ($I_{disp}$) was then multiplied by a term $(1-10^{-\eta/10})$ to factor out the photocurrent due to the light that was coupled into the waveguide at the optimal alignment, where $\eta$ is the single-facet coupling loss of the edge coupler (in dB). Here it is assumed that the 3 μm fiber displacement resulted in no change in the absorption of the uncoupled light into the PD. The obtained value was then subtracted from the $I_{ph}$ to get the estimated effective photocurrent ($I_{eph}$) due to the light coupled into the waveguide. The above description can be expressed by

$$I_{eph} = I_{ph} - I_{disp}\left(1 - 10^{-\frac{\eta}{10}}\right). \qquad (1)$$

In future designs, the input and output waveguides can be displaced from each other to eliminate the influence of stray light.

### OE bandwidth characterization setup

To measure the PD OE bandwidth, a train of optical pulses ($\lambda = 405$ nm, full-width-half-maximum (FWHM) < 50 ps) at a repetition rate of 10 MHz from a picosecond laser diode (PicoQuant LDH-D-C-405 laser

head with PDL 800-D driver) was coupled into the PDs. The resultant electrical impulse response from a PD was captured on a 13-GHz real-time oscilloscope (Keysight Infiniium UXR0134A) connected to an RF probe (GS configuration, GGB Industries Picoprobe 40A), bias-tee (API Technologies, Inmet 8810KMF1-40, 25 V, up to 40 GHz) and a 40-GHz RF cable. We did not target a specific input polarization of the light for this measurement. The maximum optical power from the laser was 0.72 mW, and no nonlinear effects were observed.

### Capacitance and contact resistance measurements

The device capacitance was measured using an impedance analyzer (Keysight E4990A, with a 42941A impedance probe) via frequency sweeps from 20 Hz to 1 MHz. The frequency range is commonly employed for the capacitance measurement of semiconductor PN junctions[40]. The same electrical probes used for the $I-V$ measurements were connected for device probing. Standard equipment calibration, followed by fixture (i.e., electrical cables and probes) compensation (at "OPEN") was performed before the measurement. The capacitance of devices at different lengths and reverse bias voltages were measured to extract the per-length junction and parasitic capacitance. The measurements are shown in the Supplementary Information. The parasitic capacitance is found to be $70 \pm 21$ fF and $55 \pm 17$ fF for PIN and PN devices, respectively, throughout the measured reverse biases.

The contact resistance was extracted from the device $I-V$ characteristics at forward bias. In an ideal forward-biased PN junction, the current increases exponentially with voltage. In practice, the current switches to a linear increase beyond a certain forward-bias voltage, when an external resistance (e.g., contact resistance) becomes more significant than the junction forward resistance.

From the extracted capacitance and resistance, and accounting for 50 Ω load resistance from the measurement apparatus, the calculated $RC$-limited 3-dB OE bandwidth for the 50-µm long devices approaches 30 GHz at $|V_r| \sim 10$ V for both PN and PIN configurations (see Supplementary Information), significantly higher than the measured results, which were limited by the laser pulse width and the instrumentation.

### Definition of APD parameters

The avalanche multiplication gain, $M$, is given by

$$M(V_r) = \frac{I_{ph}(V_r) - I_{dark}(V_r)}{I_{ph}(V_{M=1}) - I_{dark}(V_{M=1})}, \tag{2}$$

where $I_{dark}$ is the dark current, $V_r$ is the reverse bias voltage, and $V_{M=1}$ is the bias voltage at $M=1$. The avalanche breakdown voltages $V_{br}$ can be extracted from[41]

$$\left(\frac{V_r}{V_{br}}\right)^n = 1 - \frac{1}{M(V_r)}, \tag{3}$$

via linear interpolation, where $n$ is an empirical parameter. $V_{br}$ was found to be $-13.3 \pm 0.9$ V and $-13.8 \pm 1.1$ V, respectively, for 50 and 100 µm-long devices (see Supplementary Information). As $V_{br}$ varied among devices, we used $|V_r|/|V_{br}|$ to compare $M$ and the 3-dB OE bandwidth of different devices. The bandwidth measurement was identical to that for Fig. 2e, except we held $P_{in,ref} = -13.0 \pm 0.5$ dBm, the same as that in the avalanche gain measurement. As $V_r$ approached $V_{br}$, $M$ substantially increased, while the 3-dB bandwidth decreased. At $V_r = V_{br}$, $M = 46 \pm 14$ and $29 \pm 9$, respectively, for the 50- and 100-µm long devices. Correspondingly, the 3-dB bandwidth reduced from $7.6 \pm 0.3$ to $4.0 \pm 1.4$ and $3.7 \pm 1.6$ GHz. The influence of the input pulse width (estimated 3-dB OE bandwidth ~ 9 GHz) and the oscilloscope bandwidth (13 GHz) on the device 3-dB bandwidth can thus be considered negligible at $V_{br}$. Similar bandwidth reductions have been observed in other works[33,42–45] and can be explained by an increased

avalanche build-up time at higher $M$. We further calculated gain-bandwidth product (GBP) up to $M(V_{br})$. At $V_r = V_{br}$, GBP = $173 \pm 30$ and $99 \pm 15$ GHz, respectively, or 50- and 100-µm long devices. The values are reasonable compared to[33,46], indicating good APD performance. The $M$, 3-dB OE bandwidth and GBP results for the 100-µm long devices can be found in the Supplementary Information (Fig. S17).

The avalanche breakdown voltage $V_{br}$ was determined using Eq. (3), which is equivalent to

$$\ln\left(1 - \frac{1}{M(V_r)}\right) = n \ln(|V_r|) - n \ln(|V_{br}|). \tag{4}$$

Therefore, performing a linear fitting and extrapolation between $\ln\left(1 - \frac{1}{M(V_r)}\right)$ and $\ln(|V_r|)$ leads to the extraction of $n$ and $n \ln(|V_{br}|)$, via its slope, $a$, and y-intercept, $b$, respectively. $V_{br}$ can thus be calculated using $|V_{br}| = \exp(-b/a)$. The Supplementary Information (Fig. S18) shows an example plot of the fitting for a 50-µm long PN device, resulting in a $V_{br} = -13.1$ V. $V_{br}$ of other device lengths were similarly determined using this approach.

### Wavelength-tunable microring filter measurement

The photonic integrated circuit comprises a SiN racetrack microring resonator (radius = 28 µm, straight arm length = 12 µm) with a TiN resistive heater above, and a PIN photodiode (200 µm-long) coupled from the bus waveguide (at the through port) of the ring via a directional coupler. The measurement setup is similar to the DC characterization setup described above, except we used two additional electrical probes (MPI MP40) to power the heater via the other channel of the Keysight B2912A SMU. TE-polarized light at $\lambda = 514$ nm from the Coherent OBIS Galaxy laser source was coupled into the bus waveguide and both the photocurrent from the PD and the transmitted optical power at the through port, $P_{thru}$, were recorded from the SMU via script control, as a function of the applied electrical power to the heater.

## Data availability

Data underlying the results presented in this paper are available at https://doi.org/10.17617/3.U7YTAS.

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

## Acknowledgements

The authors thank J. Groth for assistance with data analysis, A. Stalma-shonak for assistance in the measurements. The authors are grateful for the loan of the picosecond laser from PicoQuant GmbH. J.K.S.P. acknowledges support from the Natural Sciences and Engineering Research Council of Canada (grant number: RGPIN-2018-06491).

## Author contributions

W.D.S., J.C.C.M., and J.K.S.P. conceived the initial idea. Y.L., A.S., and J.M. performed the electromagnetic simulations, and Z.Y. completed the TCAD simulations. The layout was generated by Z.Y. and W.D.S. X.L. and P.G.Q.L. were responsible for device fabrication. S.S.A. designed the PIC with microrings and PDs. Y.L. and H.C. carried out the measurements. Y.L., J.K.S.P., and J.M. analyzed the data. J.K.S.P. and Y.L. co-wrote the manuscript with inputs from other co-authors. All work was done under the supervision of J.K.S.P.

## Funding

## Competing interests

The authors declare no competing interests.

## Additional information

**Correspondence and requests** for materials should be addressed to Yiding Lin or Joyce K. S. Poon.

