## [Peer review file · Nature Communications]

REVIEWER COMMENTS

Reviewer #1 (Remarks to the Author):

The paper titled "Monolithically integrated, broadband, high-efficiency silicon nitride-on-silicon waveguide photodetectors in a visible-light integrated photonics platform" provides noteworthy results and it has a concrete significance in the field of photonic integration.

Congratulations to the authors for writing a well-explained paper. I have a few suggestions to enhance the impact of this great work before publication.

1. The authors have made a few design choices. They gave sufficient explanations for those design choices. But they did not provide enough justification for the precise numbers they have decided. Example: The placement of metals and vias to avoid any loss in the SiN waveguide. It can be helpful to show some study showing the safe distance? Same for the Si mesa height of 2.85 μm for low propagation loss. Showing a trend about how Si mesa height impacts the loss can be helpful for the future research work.
2. Fig 1 shows a lot of dimensions. Some dimensions referenced in the text do not appear in the figure. For example, the width of the intrinsic region had a width of 2 μm .
3. It is good that the authors mention their earlier work for some details. In some cases, they can provide the relevant numbers. Example: Page 2 line 114: propagation loss and insertion loss of edge-couplers. The same holds for Fig. S1. Simply, give a one-liner about the used procedures. For more details, the reader can always go to the reference. More generally, the manuscript should be complete in its presentation.
4. Fig. S1: For some waveguide losses, there is an error of ± 0.4 dB/cm and for some ± 0.8 dB/cm. Why that?
5. What is the reason for wavelength dependence for the coupling loss? And why a difference between TE and TM modes?
6. Can it be verified that the EQE discrepancy at 405nm (TM mode) is due to fabrication imperfections? What are the factors that the authors eliminated before reaching to this conclusion.
7. Fig. S6 has some arrows missing.
8. Some references show "???".
9. It is better to provide quantified numbers. I read words like "near-optimal", "marginal change", and "reasonable EQE". They are not helpful. Please make the statements more quantified wherever possible.
10. Were longer than 50 μm devices also tested? If yes, then what was the experimental outcome. And also, was the robustness of the EQE to device dimensions verified by experiments as well. What was the matching for the simulation vs experimental results in this case?

11. Give the color code for the voltage V_r in figure 2(e).
12. It is understood from the paper that the PDs were fabricated on a 200mm wafer. How was the performance of the PDs over the whole wafer?
13. As the heater power increases in Fig. 3, the through port loss increases. Any explanation for that?
14. Can it be tabulated that how the performance of the end-fired coupled SiN-on-Si PD / MSM is w.r.t to the current work? It will solidify the claim of "a new state-of-the-art".
15. What is limiting the bandwidth of these modulators?
16. The authors use Palik dispersion model for Si in simulation. They use a fixed refractive index at a wavelength for SiN and SiO₂. Why? Probably this simulation will be accurate if the dispersion is used for all materials.
17. Is there any impact of loss due to modal mismatch on the detector performance such as the EQE?
18. The quality of Figure S4 can be improved. It seems that the two blue curves are perfectly overlapping for TM mode EQE. Why that and why not for TE mode?
19. Many figures have limited description in the caption. The authors should put more effort into briefly elaborating the key findings in the caption.

Reviewer #2 (Remarks to the Author):

This is a great paper that presents a high-performance waveguide integrated visible light photodetector. The device works at GHz speeds over a large optical bandwidth of several hundred nm, has a high quantum efficiency, and is compatible with a standard fabrication platform on silicon. Visible light photonics is becoming increasingly important for a number of applications (quantum and atomic physics, biology and chemistry, display technology, etc) and active devices for the visible have been an underexplored, and very much needed, area of integrated photonics. The large optical bandwidth, while expected, is an especially nice feature of this device. This paper is also valuable for its particularly thorough experimental characterization and theoretical exploration of the proposed device. It looks at a variety of wavelength and geometries and both polarization states. Since visible light photonics applications often need a variety of wavelengths and more than just the standard TE polarization, this broader view is very useful to the designer and I don't believe has been presented before.

The work presented in this paper is a significant extension the authors' CLEO paper from earlier this year (which only presented DC data for blue light in one polarization for PIN devices). The authors should cite this previous paper in the text (they don't currently).

Before publication, it would be good for the authors to address the following, mostly minor, points:

1. In regards to the specific claims of novelty

a. The authors give a good survey of the literature. The most relevant paper being the Yanikgonul et al. paper published in this journal last year, which presents another waveguide integrated visible light photodetector, but one that device is only measured at one wavelength, and requires a much more complicated, less practical fabrication method. The authors do appear to have missed one other relevant paper, from our group:

“R. Morgan, D. Kharas, J. Knecht, P. Juodawlkis, K. Cahoy and C. Sorace-Agaskar, "Waveguide-Integrated Blue Light Detector," 2021 IEEE Photonics Conference (IPC), 2021”

Which they should consider citing. Our paper also uses a standard foundry process to make a blue-light detector, but the detector doesn't have the same wide optical bandwidth as the one presented here and speed and avalanche operation data is not presented.

b. While, I think that this paper is an important addition to the literature, and represents a meaningful step forward in terms of achieving a practically usable device, I think this claim in the introduction:

“we report the first broadband, high-efficiency, SiN-on-Si waveguide PDs integrated within a foundry fabricated visible spectrum PIC platform fabricated on 200mm Si wafers”

should be modified to assert that it was the first within “a standard open access foundry visible spectrum PIC platform...” to more clearly differentiate it from other recent papers in the published literature. The corresponding claim in the abstract:

“ the first waveguide photodetectors that are monolithically integrated in a visible light photonic platform on silicon”

should also be narrowed accordingly.

2. One of the strengths of the paper is the many different measurements that are done (multiple wavelengths, multiple lengths and geometries, TE and TM polarization, PIN and PN devices).

a. However, it is not always specified which measurements correspond to what, and, when it is specified, it isn't always easy to find. For example, it's not clear at what polarization the OE bandwidth was measured, nor is the polarization of the measurement at location 136-138 (“At $\lambda = 488$ nm, photocurrent of $\sim 1.35 \mu\text{A}$ was observed... throughout the V_r range”) clear (though one can do some math and compare to the graphs to figure it out). Similarly, it is not immediately clear which EQE measurements correspond to the PN device (or are these not in the paper?). The authors should try to go through and make sure that all the variables are clearly specified for each result. One or more tables summarizing the measurements for a specified geometry or set of geometries could be very helpful here.

b. Similarly, it's not always clear which measurements are on the same device and which are taken on different ones. Specifically, are all the pictures in figure 2.a. the same device or a different one for each

wavelength? If it is the same device, is it the exact same device or copies hooked up to different input routing waveguides? If they are different devices, what is the optical bandwidth of the individual device? Are pictures 2b-e all on the same device? And is it the same one used for the pin measurement in 2f?

3. Regarding photodiode efficiency:

a. The way the authors calibrate out the extra light hitting on the photodiode (I_{disp}) is acceptable, but non-ideal, as it is prone to error. The more standard way of handling this is to introduce a jog in the input waveguide to offset the photodetector out of the input fiber beam. Likewise, the comment (loc 331) that this can be avoided in the future by having the input and output at 90 degrees, suggests a needlessly complex solution and should be changed.

b. The EQE simulations in S4 appear to be independent of whether the devices simulated is a PN or PIN device. Do the authors expect this to matter? What fraction of the light is expected to be absorbed outside of the depletion region in both cases and how should this affect the device efficiency?

c. Given this and the clear presence of back reflections in the FDTD simulations of Fig S3 (at least at longer wavelengths). I have a little trouble believing the authors' implication that the only loss in efficiency comes from mode overlap. It would be good if these other things were somewhat quantified via simulation and shown to be negligible.

d. The authors claim that the decrease in 405 nm responsivity may be due to fabrication imperfection (loc 171). This doesn't follow for me and should be changed or clarified. What fabrication imperfection would cause this and not affect the other wavelengths measured (or were these other wavelengths measured on different devices)? Further, I noticed that the waveguide loss was much higher at 405 nm (10 dB/cm), and that loss measurements appear to be from the same "wafer run" and not necessarily the same wafer or die. Material loss in SiN at these low wavelengths ($< \sim 430$ nm) can be highly variable (they may have statistics on this specific to their process), which would affect the calibration and could also account for the discrepancy.

4. The paper notes that this platform also supports suspended heater structures as a standard design, but the heater used in the ring demonstration at the end is not suspended. The authors should clarify why not (is there something about the photodiode design or fabrication that prevents this?) and whether other suspended structures were present in other parts of the wafer (if not, why not?).

5. I'm not sure that the graph in Fig S9 supports the idea that a tsi of 2.85 μm is truly sufficient – oscillations are still visible at this height for wavelengths $> \sim 500$ nm. The authors should change or clarify this.

6. Similarly, given that I would expect the longer wavelengths to be more sensitive to changes in W_{gn} and W_{si} , it would be nice if Fig S9 c and d included lines for at least one wavelength longer than 488 nm.

Also, why were different device lengths used in each of these simulations? This should be made consistent or justified.

7. For the calculations shown in figure S10, it would be good to know what the EQE curve looks like for longer length (given that the tradeoff here is likely to be between devices of longer length and larger W_{gap} and those with close W_{gap} and shorter length). The tradeoff with length and speed (if meaningful) would then be nice to know to allow for full device design.

8. There are a number of small typos and the like that should be fixed:

a. In figure 1A the “SiN 1” label is somewhat confusingly placed and should be moved to be closer to the bottom layer

b. Figure S4 should be redone to make clearer which line corresponds to which length and that two of the blue lines are on top of each other

c. Loc 54 in the supplemental material: ?? -> 5

d. The dashed curves are not visible in figure S7. Also, it would be nice if the maximum possible responsivity was noted in the figures.

e. This is a bit nit-picky, but it would be nice to see the whole forward bias curve in Fig S13 and not just the linear part.

f. It would be helpful to include the APD OE bandwidth in the section of the main paper that starts on line 201. This prevents the reader from having to hunt for it.

g. The refractive indices are given, but it is not specified how they were measured. This should be added (Loc 285).

The following things would make the paper stronger, but (in my opinion) do NOT need to be addressed to allow for publication:

1. The measurement and analysis of the high-speed and APD performance of the device was less thorough than the rest of the paper. This is to be expected as these measurements are more complex and often limited by available equipment.

a. It would be nice if the speed could be characterized in both polarizations or for more than just 405 nm light. If this is not possible, some discussion on whether the speed would be expected to vary with polarization (presumably not) or wavelength (mildly?) would add to the paper.

b. Similarly, the paper would be strengthened if both the linearity and speed data had been taken at the same wavelength to provide a complete characterization at one wavelength. I assume that this wasn't done due to instrument limitations.

c. Given that the material parameters of silicon are very well understood, it should be straight forward to simulate the expected carrier transit times in, say, Sentaurus for this device and determine if they match the measured speeds at low bias (before the measurement system becomes the limiting factor or avalanching starts (where simulation gets more tricky)). I'm not sure why this wasn't done.

d. The APD characterization could include the excess noise figure, but is also good as is.

2. I agree with the authors that a dopant well structure can be used to help suppress the device cross-talk, but think the picture they have presented in section S12 is a overly simplistic, and I'm not sure it's adding anything. I would consider either cutting section S12 and leaving the comment at loc 265 to stand on its own, or expanding it (though I don't think either is necessary for publication).

On the whole, I liked this paper and support its publication in Nature Communications.

Reply to the Reviewers

General comments

During the revision of the manuscript, we discovered an error in the boundary settings of the FDTD simulations in the original manuscript. We have redone the simulations and have included details on the simulation settings in Section S5 of the Supplementary Information. Therefore, you will find that some simulated values have changed compared to our original submission; however, the main conclusions of this work remain the same.

The revised portions of the manuscript are highlighted in yellow. Additional edits for clarity aside from those in response to the reviewers' questions are also highlighted.

Reviewer #1:

The paper titled "Monolithically integrated, broadband, high-efficiency silicon nitride-on-silicon waveguide photodetectors in a visible-light integrated photonics platform" provides noteworthy results and it has a concrete significance in the field of photonic integration.

Reply: We thank the positive evaluation and the detailed review from the reviewer.

Congratulations to the authors for writing a well-explained paper. I have a few suggestions to enhance the impact of this great work before publication.

1. The authors have made a few design choices. They gave sufficient explanations for those design choices. But they did not provide enough justification for the precise numbers they have decided. Example: The placement of metals and vias to avoid any loss in the SiN waveguide. It can be helpful to show some study showing the safe distance? Same for the Si mesa height of 2.85 μm for low propagation loss. Showing a trend about how Si mesa height impacts the loss can be helpful for the future research work.

Reply: Thank you for your comments. We computed the SiN waveguide loss using a finite difference eigenmode (FDE) solver (Lumerical MODE Solutions) while making these design decisions. We have now added Section S1 and Figure S1 to show these loss calculations.

Figure S1a shows the results for both TM_0 and TE_0 modes at 4 visible wavelengths (405, 488, 532 and 640 nm). In Fig. S1(a), the waveguide loss is < 0.02 dB/cm for all the wavelengths at a SiN-Via1 gap larger than 6 μm . Thus, the SiN-Via1 distance was set to 6 μm . The inset in Fig. S1(a) shows the schematics for the SiN-Via1 parameters used in the calculation. The Si substrate was not included to extract the influence from the Al via. The background index was 1.46. The simulation space used metal boundary conditions. At the boundary, for the wavelength of 640 nm (least localized mode), the electric field magnitude has dropped to 10^{-7} of the maximum.

Figure S1(b) shows the SiN waveguide loss as a function of Si mesa height. Here the gap between SiN and Si (W_{gap}) was 190 nm, and the SiN thickness (t) was 120 nm. The waveguide widths (W_{gw}) for the calculation are 380 nm at $\lambda = 405, 488$ and 532 nm, and 500 nm at $\lambda = 640$ nm. From the calculation, for $W_{\text{gw}} = 380$ nm, the loss is $< 10^{-6}$ dB/cm at the mesa height of 2.85 μm for wavelengths up to 532 nm. For longer wavelengths, the loss can be less than 4×10^{-4} dB/cm for $W_{\text{gw}} = 500$ nm up to $\lambda = 640$ nm. Therefore, Si mesa height is set to 2.85 μm with $W_{\text{gw}} = 500$ nm for a low propagation loss up to $\lambda = 640$ nm, and we used $\lambda = 514$ nm for the tunable microring circuit with $W_{\text{gw}} = 380$ nm.

2. Fig 1 shows a lot of dimensions. Some dimensions referenced in the text do not appear in the figure. For example, the width of the intrinsic region had a width of 2 μm .

Reply: Thanks for the comment. Due to the space limit in the figure, we have included the description in the caption of Fig. 1(a). We have also labelled the i -region in Fig. 1(d). The amended caption is:

The cross-section of the PDs, which has a PN (shown in (a)) or PIN junction (intrinsic region width = 2 μm , a top view as shown in (d)), is demarcated by the red dashed box.

3. It is good that the authors mention their earlier work for some details. In some cases, they can provide the relevant numbers. Example: Page 2 line 114: propagation loss and insertion loss of edge-couplers. The same holds for Fig. S1. Simply, give a one-liner about the used procedures. For more details, the reader can always go to the reference. More generally, the manuscript should be complete in its presentation.

Reply: Thank you for the suggestion. We have provided the relevant numbers as follows regarding earlier works:

Page 2, line 116:

We have previously reported the propagation loss of the routing waveguides (~ 1.1 -5.9 dB/cm from 430-648 nm [2], and ~ 1.8 -7.1 dB/cm from 405-640 nm [32]) and insertion loss of the edge couplers (7.5-11.3 dB/facet from 430-648 nm [2], and 5.9-8.6 dB/facet from 405-640 nm [32]).

Regarding Fig. S2 (in revised Supplementary Information), we have added the following in Section S2 for completeness:

Cleaved single-mode fibers were used to couple light into and out of the respective test structures on chip [1,2].

4. Fig. S1: For some waveguide losses, there is an error of ± 0.4 dB/cm and for some ± 0.8 dB/cm. Why that?

Reply: The error is due to the variability in the input/output coupling loss and the waveguide loss. In Fig. S2 of our earlier work [Lin *et al.*, Opt. Express 29, 34565-34576 (2021)], we have also observed similar loss errors for PECVD SiN waveguides on a separate wafer run. For example, at $\lambda = 532$ nm, propagation losses were 2.8 ± 0.3 and 4.3 ± 0.7 dB/cm for waveguides (150 nm thick) with widths of 520 and 380 nm, respectively.

For clarification, we have added the explanation below in the revised Supplementary Information (Section S2, highlighted in yellow):

The measurement error is due to the variability in the input/output coupling efficiency and the waveguide loss. We have also observed similar loss variability on the wafers in [2].

5. What is the reason for wavelength dependence for the coupling loss? And why a difference between TE and TM modes?

Reply: The higher coupling loss at shorter wavelengths is due to the reduced coupling efficiency between the fiber and the edge coupler [Fig. S3 (a) of Lin *et al.*, Opt. Express 29, 34565-34576 (2021)]. The mode field diameter becomes smaller at both the fiber and coupler tip at shorter wavelengths, which leads to the reduced coupling efficiency under the same fiber-waveguide alignment precision.

The lower coupling loss for TM than TE is due to the lower optical confinement of the fundamental TM mode and correspondingly a better overlap with the fiber mode. Similar losses were observed in Fig. 3 of Sacher *et al.*, Opt. Express 27, 37400-37418 (2019).

For clarification, we have added the explanation below in the revised Supplementary Information (Section S2, highlighted in yellow):

The higher coupling loss at shorter wavelengths is due to the compromised coupling efficiency between the fiber and the edge coupler [2]. The lower coupling loss for TM than TE was due to the lower optical confinement of the fundamental TM mode in the coupler and correspondingly a better mode overlap with the fiber [1].

6. Can it be verified that the EQE discrepancy at 405nm (TM mode) is due to fabrication imperfections? What are the factors that the authors eliminated before reaching this conclusion?

Reply: In the experiment, we measured the same devices on 3 different chips distant from each other (>10 cm) on the wafer and the 3 measurements had similar EQE (Fig. 2(d)). With the new FDTD simulation results, we find that the simulated EQE is higher than the measured EQE for nearly all wavelengths, and the deviation is most significant at wavelengths near 532 and 594 nm.

Our FDTD simulations assumed the SiN waveguide to be lossless, but even for a SiN propagation loss of 10 to 20 dB/cm (as expected at 405 nm in the worst case), for a 50 μ m length, less than ~2% of the light would be dissipated to waveguide propagation loss. Introducing an imaginary refractive index to the SiN equivalent to

~10dB/cm of propagation loss ($k = 7.42 \times 10^{-6}$) made a negligible difference in the absorbed power in the Si.

The difference between the simulations and experiments can be explained by several causes. First, the simulations assume a perfect internal quantum efficiency for all wavelengths. In practice, the internal quantum efficiency is reduced by surface recombination and is wavelength-dependent, since longer wavelengths generate carriers in deeper regions of the Si (i.e., beyond 1 μm depths), and those carriers may not be as efficiently collected.

Second, for wavelengths near and longer than 500 nm, the modes are near cut-off, so the EQE is sensitive to W_{gn} (see Fig. S11(c)). A narrower than expected W_{gn} would have led to a reduction of EQE.

Third, for short wavelengths, the routing waveguides (with widths of 500 nm for the single PD devices) are multimode (supporting 2 TE and 2 TM modes at $\lambda=405$ nm), while the narrow SiN waveguides over Si mesa are single-mode (supporting only the TE_0 and TM_0 modes). Power in the higher order mode would be radiated away in the taper transitions, leading to excess loss. However, we cannot directly confirm whether the higher order mode was excited or how much power could have been coupled into higher order modes (e.g., by waveguide sidewall roughness or slanted waveguide walls).

We have revised the explanation of the EQE to the following:

The measured EQE qualitatively agrees with but is lower than the simulated results, which assume an ideal internal quantum efficiency and lossless SiN waveguides and transitions (see Supplementary Information).

...

The measured EQE is limited by a number of factors, such as the internal quantum efficiency, mode mismatch loss between the input SiN waveguide and the SiN-on-Si region, W_{gn} variations, device length, scattering and back-reflection at the Si mesa interface, and excitation of high order modes in the routing waveguide at short wavelengths (e.g., near 405 nm) (Supplementary Information). Nonetheless, Fig. S7 (Supplementary Information), shows a device length of $l = 50 \mu\text{m}$ is sufficient to saturate the EQE for $\lambda > 445$ nm.

Because the EQE is affected by numerous factors, we have removed the mode overlap curve, η_{mode} , in Fig. 2(d) to avoid confusion. The discussion of the mode overlap remains in the Supplementary Information section.

7.Fig. S6 has some arrows missing.

Reply: Thanks for pointing this out. The arrows were not placed in the 405- and 445-nm figures, because the scattering into Si was barely observable. As the study had revealed the stronger scattering into Si with an increasing wavelength, we thus decided not to include the arrows in the 405- and 445-nm figures. For clarification, we have added the below sentences in the caption of the current Fig. S8:

The arrows were not included in the figures at 405 and 445 nm due to weak scattering.

8. Some references show “???”.

Reply: Thank you for pointing this out. We have carefully checked the manuscript and changed all “???” to the corresponding numbers.

9. It is better to provide quantified numbers. I read words like “near-optimal”, “marginal change”, and “reasonable EQE”. They are not helpful. Please make the statements more quantified wherever possible.

Reply: Thanks for the comments. We have amended these sentences as follows:

1. **Main manuscript page 3, line 181:** ...We verified that longer devices (up to $l = 500 \mu\text{m}$) showed an increase of $< 30\%$ in R .

30% came from the responsivity increase for 500- μm devices compared to the 50- μm devices in Fig. S9.

2. **Main manuscript page 3, line 171:** ...the EQE benefits from the absorption of scattered light (Supplementary Information, Fig. S8).

10. Were longer than 50 μm devices also tested? If yes, then what was the experimental outcome. And also, was the robustness of the EQE to device dimensions verified by experiments as well. What was the matching for the simulation vs experimental results in this case?

Reply: Thanks for the questions. Yes. We have also tested devices with lengths up to 500 μm . The results are shown in Fig. S9. Unfortunately, we do not have the experimental results for the EQE robustness to device dimensions at the moment. The comparison between the simulation and the experimental results can be planned for future work.

11. Give the color code for the voltage V_r in figure 2(e).

Reply: Thanks for the comment. Due to the space limit in the figure, we have moved the legend in Fig. 2(e) to its caption. To avoid future confusion, we indicated in the main text (and highlighted in yellow) that the legend can be found in the figure caption.

Main manuscript page 3, line 196:

Figure 2(e) displays the measured impulse response of a device at different V_r (the line labels are in the caption).

12. It is understood from the paper that the PDs were fabricated on a 200mm wafer. How was the performance of the PDs over the whole wafer?

Reply: Thanks for the question. For the optical (EQE and gain) and high-speed device performance, we tested 3 chips that were > 10 cm from each other on the wafer. The error bars/overlapping plot of the corresponding data (Fig. 2(d) and (g)) indicated the performance uniformity of the devices over the whole wafer. We can see that the EQE and GBP variations are within $\pm \sim 30\%$ of the mean values.

For the dark current characterization, we tested more devices from 7 chips across the wafer. We also observed $\pm 25\sim 30\%$ variation in the collected data (i.e., 144 ± 42 and 266 ± 65 pA at -5 and -15 V, see page 2 of the main text in the section on PIN Photodiodes).

These variations are comparable to that ($R = 0.65\pm 0.18$ A/W and GBP = 234 ± 25 GHz) reported in Yanikgonul, S., Leong, V., Ong, J.R. *et al.* Integrated avalanche photodetectors for visible light. *Nat Commun* **12**, 1834 (2021). <https://doi.org/10.1038/s41467-021-22046-x>, which was manufactured in the same foundry.

We have amended the following discussion in the “Discussion and Conclusion” part of the manuscript:

Compared to the Si PD end-fired coupled to a SiN waveguide in [33], which was characterized only at 685 nm, our SiN-on-Si PDs exhibited a higher EQE across a significantly wider wavelength span while maintaining a similar GBP (173 ± 30 GHz in this work vs. 234 ± 25 GHz in [33]) and wafer-level performance uniformity (up to $\pm \sim 30\%$ variation on dark current, EQE, M , OE bandwidth and GBP).

13. As the heater power increases in Fig. 3, the through port loss increases. Any explanation for that?

Reply: Thanks for the question. We believe the through port loss was due to a slight mechanical drift of the chip caused by the heating leading to the fiber coupling misalignment. In contrast, the monitored PD photocurrent did not show a loss increase. We thus suspect the fiber misalignment was mainly from the output fiber at the through port.

We have added the following statement in the manuscript:

The slight increase in the insertion loss observed at the through port with the increase of the heater power was likely due to a drift of the output fiber coupling. The insertion loss increase was not observed by the on-chip PD.

14. Can it be tabulated that how the performance of the end-fired coupled SiN-on-Si PD / MSM is w.r.t to the current work? It will solidify the claim of “a new state-of-the-art”.

Reply: Thank you for the comment. The table below summarizes the performance of recent visible light PDs. The table has been added to Supplementary Information Section S13.

Table R1. Performance comparison of SiN waveguide-integrated visible and NIR ($\lambda = 400\text{-}800\text{ nm}$) PDs.

Type	A (μm^2)	λ (nm)	I_{dark} (pA)	EQE (%)	BW (GHz)	M	GBP (GHz)	Ref.
SiN-on-Si mesa (p-i-n , p-n)	24×50	400-640	144@-5V* 266@-15V*	60-88 @-2V*	8.6±1.0 @ -20V*	46±14†	173±30 †	This work
End-fire SiN-on-SOI (p-n)	6 ^a ×16	685	<70@-2V	~40 ^a	30	12.3	234±25	[R1]
SOI-on-SiN (p-i-n)	11.6×200	775, 800	107@-3V	~30	6	~10 ^a	68	[R2]
Al ₂ O ₃ -on-SOI (p-i-n)	N.A.×100	405	<1000	76	-	-	-	[R3]
poly-Si (MSM)	1.14×10	654	200@-5V	67 ^a	-	-	-	[R4]
a -Si (MSM)	30 ^a ×50	660	25@4V, 50@8V	0.06 ^a	1×10 ⁻⁶	-	-	[R5]
MoSe ₂ / WS ₂ -on-SiN	5 ^a ×13	780	50	158@- 2V ^a	0.02	-	-	[R6]
AlGaAs/ GaAs-on-Ta ₂ O ₅	20×20 20×40	635	20@-2V	22.4	12.6	-	-	[R7]

Legend: A : device active area (width \times length); λ : operating wavelength range; I_{dark} : dark current; BW: 3-dB OE bandwidth at unity/low gain; *data from PIN devices; †data from PN devices; N.A.: not available. ^aThe results were not explicitly reported but inferred from relevant data in literature.

We can see from the table that our devices have the broadest wavelength range demonstrated to date while exhibiting a competitive dark current and APD gain-bandwidth product compared to other designs. Our work enables PICs requiring broadband high-efficiency PDs and APDs at both visible and NIR wavelengths. In the discussion section of the manuscript, the comparison was summarized.

15. What is limiting the bandwidth of these modulators?

Reply: Thanks for the question. We believe the reviewer is asking about the bandwidth limiting factors for the PDs. As seen in Fig. S15, the measured 3-dB bandwidth for PIN devices was limited by carrier transit time at a low bias ($|V_r| < 10$ V), and the laser pulse width and the measurement instrumentation (oscilloscope) at a high bias ($|V_r| > 10$ V). For the PN devices, the 3-dB bandwidth was limited by the instrumentation for $|V_r| > 1$ V. In the event of the avalanche effect, the bandwidth was limited by the avalanche build-up time. This has also been discussed in the main manuscript (page 4, at the end of paragraph 1) and Supplementary S10C.

16. The authors use Palik dispersion model for Si in simulation. They use a fixed refractive index at a wavelength for SiN and SiO₂. Why? Probably the simulation will be accurate if the dispersion is used for all materials.

Reply: Thanks for the question. We used the Si-Palik model for the Si material because the refractive index of Si changes significantly ($3.78 + 0.013i$ to $5.57 + 0.387i$) at the wavelengths of interest (400-700 nm). Our ellipsometry measurements show that the index dispersion for SiN and SiO₂ was lower (SiN~1.78-1.83 and SiO₂~1.45-1.46). Therefore, we used a fixed index for SiN (1.82) and SiO₂ (1.46) in the simulation. For the PD designs, the slight variation of the SiN and SiO₂ refractive indices do not affect the results.

17. Is there any impact of loss due to modal mismatch on the detector performance such as the EQE?

Reply: Thanks for the question. The impact of mode mismatch on EQE has been accounted for in the simulated EQE results in Fig. 2(d). At short wavelengths, the mode mismatch loss is lower, but the high mode confinement in the SiN necessitates a longer PD length for a high EQE. At longer wavelengths (> 532 nm), despite the higher mode mismatch loss, a high EQE is possible due to the absorption of the scattered light (Fig. S8).

18. The quality of Figure S4 can be improved. It seems that the two blue curves are perfectly overlapping for TM mode EQE. Why that and why not for TE mode?

Reply: Thanks for the comment. The TM curves overlap because the EQE has reached a maximum at $l = 20 \mu\text{m}$. This is not seen for TE because the coupling rate from the SiN waveguide into Si is higher for TM than TE. From Table S1, the evanescent coupling coefficient, α_{coupling} , is ~ 4 to 5 times larger for TM than TE for $\lambda = 405$ to 532 nm. **To improve clarity, Fig. S6 has been revised.**

19. Many figures have limited description in the caption. The authors should put more effort into briefly elaborating the key findings in the caption.

Reply: Thanks for the comment. We have expanded the elaboration of key findings in the figure captions below:

Fig. S7: Simulated EQE as a function of device length l at different wavelengths. **The data points are fitted with an exponential and the extracted parameters are tabulated in Table S1.**

Fig. S8: Cross-sectional $|E|$ profiles at different visible wavelengths. The cladding is SiO_2 . **Light scattering into Si was more significant at longer wavelengths since the input mode is less confined in the SiN waveguide. The arrows are not included in the figures at $\lambda = 405$ and 445 nm due to weak scattering.**

Fig. S9: Measured responsivity of PIN devices as a function of device length (l) at several wavelengths (TM: solid curves; TE: dashed curves). Each data point was averaged from the measurements of 3 chips far apart on the wafer. **Each data point was averaged from the measurements of 3 chips far apart on the wafer. The responsivity increased by $< 30\%$ for devices with $l > 50 \mu\text{m}$.**

Fig. S10: Measured and simulated EQE for the SiN-on-Si photodiodes as a function of W_{gn} at $\lambda = 405$ nm for the TE polarized mode. **Narrowing the W_{gn} up to 150 nm enhances the evanescent light penetration into Si and consequently the EQE.**

Fig. S11: **EQE vs. Si thickness, SiN and Si width variations.** (a) Schematic of SiN-on-Si PD showing the parameters being studied. (b-d) Simulated EQE as a function of (b) Si thickness (t_{Si}) at different visible wavelengths, (c) SiN width (W_{gn}) and (d) Si mesa width (W_{Si}) at $\lambda = 488$ and 640 nm. **The device length (l) used in the simulations is $50 \mu\text{m}$. Other geometrical parameters are as follows when held constant: $W_{\text{Si}} = 2 \mu\text{m}$; $W_{\text{gn}} = 200$ nm; $t = 120$ nm; $W_{\text{gap}} = 190$ nm; and $t_{\text{Si}} = 10 \mu\text{m}$.**

Fig. S12: The effect of W_{gap} on (a) η_{mode} and (b) EQE at $\lambda = 488$ nm. The device lengths in (b) are 10, 30 and $50 \mu\text{m}$, and other design parameters are: $W_{\text{Si}} = 2 \mu\text{m}$; $W_{\text{gn}} = 200$ nm; $t = 120$ nm; and $t_{\text{Si}} = 10 \mu\text{m}$. **A W_{gap} of ~ 110 -220 nm optimizes the EQE.**

Reviewer #2

This is a great paper that presents a high-performance waveguide integrated visible light photodetector. The device works at GHz speeds over a large optical bandwidth of several hundred nm, has a high quantum efficiency, and is compatible with a standard fabrication platform on silicon. Visible light photonics is becoming increasingly important for a number of applications (quantum and atomic physics, biology and chemistry, display technology, etc) and active devices for the visible have been an underexplored, and very much needed, area of integrated photonics. The large optical bandwidth, while expected, is an especially nice feature of this device. This paper is also valuable for its particularly thorough experimental characterization and theoretical exploration of the proposed device. It looks at a variety of wavelengths and geometries and both polarization states. Since visible light photonics applications often need a variety of wavelengths and more than just the standard TE polarization, this broader view is very useful to the designer and I don't believe has been presented before.

Reply: Thank you for the positive evaluation of our manuscript.

The work presented in this paper is a significant extension of the authors' CLEO paper from earlier this year (which only presented DC data for blue light in one polarization for PIN devices). The authors should cite this previous paper in the text (they don't currently).

Reply: Thank you for the kind reminder. We have added the citation to this earlier paper to our manuscript.

Page 1, Paragraph 3:

Preliminary results of the devices have been reported in Ref. [34].

Before publication, it would be good for the authors to address the following, mostly minor, points:

1. In regards to the specific claims of novelty

a. The authors give a good survey of the literature. The most relevant paper being the Yanikgonul et al. paper published in this journal last year, which presents another waveguide integrated visible light photodetector, but one that device is only measured at one wavelength, and requires a much more complicated, less practical fabrication method. The authors do appear to have missed one other relevant paper, from our group:

"R. Morgan, D. Kharas, J. Knecht, P. Juodawlkis, K. Cahoy and C. Sorace-Agaskar, "Waveguide-Integrated Blue Light Detector," 2021 IEEE Photonics Conference (IPC), 2021"

Which they should consider citing. Our paper also uses a standard foundry process to make a blue-light detector, but the detector doesn't have the same wide optical bandwidth as the one presented here and speed and avalanche operation data is not presented.

Reply: Thank you for the suggestion. This is also an important prior art that should be cited. We apologize for missing this. We have cited this in the revised main manuscript (Ref. [31]).

b. While, I think that this paper is an important addition to the literature, and represents a meaningful step forward in terms of achieving a practically usable device, I think this claim in the introduction:

“we report the first broadband, high-efficiency, SiN-on-Si waveguide PDs integrated within a foundry fabricated visible spectrum PIC platform fabricated on 200mm Si wafers”

should be modified to assert that it was the first within “a standard open access foundry visible spectrum PIC platform...” to more clearly differentiate it from other recent papers in the published literature. The corresponding claim in the abstract:

“ the first waveguide photodetectors that are monolithically integrated in a visible light photonic platform on silicon”

should also be narrowed accordingly.

Reply: Thanks for the comment. Our platform is not yet open access, but should soon be available. We believe this is the first monolithically integrated SiN-on-Si waveguide PD (the other works are Al₂O₃-on-Si and SiN end-fired coupled into Si). We have amended the abstract claims as suggested as follows:

“the first silicon nitride-on-silicon waveguide photodetectors that are monolithically integrated in a visible light photonic platform on silicon. Owing to a leaky wave design...”

2. One of the strengths of the paper is the many different measurements that are done (multiple wavelengths, multiple lengths and geometries, TE and TM polarization, PIN and PN devices).

a. However, it is not always specified which measurements correspond to what, and, when it is specified, it isn't always easy to find. For example, it's not clear at what polarization the OE bandwidth was measured, nor is the polarization of the measurement at location 136-138 (“At $\lambda = 488$ nm, photocurrent of ~ 1.35 μ A was observed... throughout the V_r range”) clear (though one can do some math and compare the graphs to figure it out).

Reply: Thanks for the comment. For the OE bandwidth measurement, we did not target a specific polarization because the OE bandwidth is determined by electrical

parameters (i.e., carrier transit time and RC-limited time constant), independent of the polarization of the input light. The same applies to the measurement of the multiplication gain. However, we confirmed identical polarization states and input power throughout the gain measurement between the PN and PIN devices.

For clarification, we have amended the manuscript as below (Methods: OE bandwidth characterization setup):

We did not target a specific polarization of the laser for this measurement

The mode polarization is TE for the results in Fig. 2(b). We have included this information in the revised manuscript:

First, we report the PIN PDs... and **TE polarized input unless otherwise stated (Fig. 2(b-e)).**

Similarly, it is not immediately clear which EQE measurements correspond to the PN device (or are these not in the paper?). The authors should try to go through and make sure that all the variables are clearly specified for each result. One or more tables summarizing the measurements for a specified geometry or set of geometries could be very helpful here.

Reply: In the manuscript, unless specified, all the results (including the quoted supplementary figures) reported under the section “PIN Photodiodes” belong to the PIN devices (Fig. 2(b-e)); while the results under the “PN Avalanche Photodiodes” belong to the PN devices (Fig. 2(f-g)). The respective device types have also been indicated in the caption of Fig. 2. We hope this can clarify the reviewer’s concerns.

We have verified separately that, at $\lambda = 488$ nm, PIN and PN PDs exhibited similar photocurrents under the same $P_{in,PD}$ (Fig. S4). This suggests a similar collection efficiency of the photon-generated carriers in the two junction designs. We thus focused our EQE measurements at low bias for the PIN devices and reported the PN devices in linear-mode operation as APD.

b. Similarly, it’s not always clear which measurements are on the same device and which are taken on different ones. Specifically, are all the pictures in figure 2.a. the same device or a different one for each wavelength? If it is the same device, is it the exact same device or copies hooked up to different input routing waveguides? If they are different devices, what is the optical bandwidth of the individual device? Are pictures 2b-e all on the same device? And is it the same one used for the pin measurement in 2f?

Reply: Thanks for pointing this out. The pictures in Fig. 2(a) were from the same PIN device with $l = 50$ μm . They were raw images taken on the same device without post-processing.

Figure 2(b)-(e) were from PIN devices with $l = 50$ μm and actual $W_{gn} = 200 \pm 20$ nm, $t = 120 \pm 10$ nm, and $W_{gap} = 190 \pm 10$ nm. The data in (d) came from 3 devices from 3

different chips on the wafer. The data in (b), (c) and (e) came from one of the 3 devices in (d).

To clarify, we added the below texts into the main manuscript:

...unless otherwise stated (Fig. 2(b-e)).

Regarding the PIN measurement in (f), yes, it was from one of the devices in (d).

For clarification, we added the following clarifications to the main manuscript (caption of Fig. 2, highlighted in yellow):

Three devices from 3 chips far apart...

The data in (b-c) and (e) came from one of the devices in (d).

The data for the PIN device came from one of the devices in (d).

3. Regarding photodiode efficiency:

a. The way the authors calibrate out the extra light hitting on the photodiode (I_{disp}) is acceptable, but non-ideal, as it is prone to error. The more standard way of handling this is to introduce a jog in the input waveguide to offset the photodetector out of the input fiber beam. Likewise, the comment (loc 331) that this can be avoided in the future by having the input and output at 90 degrees, suggests a needlessly complex solution and should be changed.

Reply: Thanks for the comment. We simply mean that the input and output waveguide should be sufficiently displaced from each other. We have changed loc 330-331 to:

In future designs, the input and output waveguides can be displaced from each other to eliminate the influence of stray light.

b. The EQE simulations in S4 appear to be independent of whether the devices simulated is a PN or PIN device. Do the authors expect this to matter? What fraction of the light is expected to be absorbed outside of the depletion region in both cases and how should this affect the device efficiency?

Reply: Thanks for the questions. As discussed in Q2a, we did not observe significant EQE differences between PN and PIN devices in the measurements; thus, the EQE simulations (which were optical absorption simulations) were independent of the junction geometry. For clarification, we have added the sentences in the revised Supplementary Information (S5, highlighted in yellow):

The simulation results are generally higher than the measurements (Fig. 2(d)), since the simulations assume a perfect internal quantum efficiency and neglect losses. We observed negligible EQE difference between PN and PIN devices (Fig. S4).

c. Given this and the clear presence of back reflections in the FDTD simulations of Fig S3 (at least at longer wavelengths). I have a little trouble believing the authors' implication that the only loss in efficiency comes from mode overlap. It would be good if these other things were somewhat quantified via simulation and shown to be negligible.

Reply: Thanks for the comment. Indeed, the EQE degradation is not solely due to the mode mismatch loss η_{mode} . Please see our response to Reviewer 1 Q6. Other factors that contribute to the EQE are: an imperfect internal quantum efficiency; the device length (e.g., Fig. S7 and Table S1 for $\lambda = 405$ nm); scattering at the interface (Fig. S8); excitation of high order modes; and W_{gn} variation (Fig. S11(c)). Previously, we included η_{mode} in Fig. 2(d) as a reference, but for clarity, we have now removed η_{mode} from Fig. 2(d).

In the original manuscript (loc 163-168), we meant that a device $l = 50$ μm is sufficient for a high EQE. We have amended the explanation as follows (main text page 3, highlighted in yellow):

The measured EQE is limited by a number of factors, such as the internal quantum efficiency, mode mismatch loss between the input SiN waveguide and the SiN-on-Si region, W_{gn} variations, device length, scattering and back-reflection at the Si mesa interface, and excitation of high order modes in the routing waveguide at short wavelengths (e.g., near 405 nm) (Supplementary Information). Nonetheless, Fig. S7 (Supplementary Information), shows a device length of $l = 50$ μm was sufficient to saturate the EQE for $\lambda > 445$ nm.

d. The authors claim that the decrease in 405 nm responsivity may be due to fabrication imperfection (loc 171). This doesn't follow for me and should be changed or clarified. What fabrication imperfection would cause this and not affect the other wavelengths measured (or were these other wavelengths measured on different devices)? Further, I noticed that the waveguide loss was much higher at 405 nm (10 dB/cm), and that loss measurements appear to be from the same "wafer run" and not necessarily the same wafer or die. Material loss in SiN at these low wavelengths ($< \sim 430$ nm) can be highly variable (they may have statistics on this specific to their process), which would affect the calibration and could also account for the discrepancy.

Reply: Thanks for the comment. Please see our response to Reviewer 1 Q6, where we have provided a more comprehensive explanation. The waveguide losses we have observed are consistent across wafers from different projects, which have included around 10 wafers. We have added the following statement comparing the experiment and simulations.

The measured EQE qualitatively agrees with but is lower than the simulated results, which assume an ideal internal quantum efficiency and lossless SiN waveguides and transitions.

4. The paper notes that this platform also supports suspended heater structures as a standard design, but the heater used in the ring demonstration at the end is not suspended. The authors should clarify why not (is there something about the photodiode design or fabrication that prevents this?) and whether other suspended structures were present in other parts of the wafer (if not, why not?).

Reply: The suspended heaters required clearances which were risky to implement in this ring resonator; therefore, they were not used. We have added in the text:

... the microring did not use the suspended structure in [23] which required clearances around the waveguide.

5. I'm not sure that the graph in Fig S9 supports the idea that a tsi of 2.85 μm is truly sufficient – oscillations are still visible at this height for wavelengths $>\sim 500$ nm. The authors should change or clarify this.

Reply: Thanks for the comment. This figure is now Fig. S11. The 2.85 μm mesa height keeps SiN waveguide propagation loss low for long wavelengths, and the EQE is close to the maximum with this Si height. The fluctuation of the EQE is reduced due to the longer device length (50 μm) and the revised simulation.

6. Similarly, given that I would expect the longer wavelengths to be more sensitive to changes in W_{gn} and W_{si} , it would be nice if Fig S9 c and d included lines for at least one wavelength longer than 488 nm. Also, why were different device lengths used in each of these simulations? This should be made consistent or justified.

Reply: Thanks for the comment. We have added the simulation at $\lambda = 640$ nm in Fig. S11(c) and (d) in the revised Supplementary Information. Similar trends are observed at $\lambda = 640$ nm compared to $\lambda = 488$ nm. We have changed Fig. S11 (b-d) so that they all share a 50- μm length.

For clarification, we have added the statement below in the caption of Fig. S11:

The device length (l) used in the simulations was 50 μm . Other geometrical parameters are as follows when held constant: $W_{\text{Si}} = 2$ μm ; $W_{\text{gn}} = 200$ nm; $t = 120$ nm; $W_{\text{gap}} = 190$ nm; and $t_{\text{Si}} = 5$ μm for $\lambda \leq 532$ nm and 9 μm for $\lambda > 532$ nm.

7. For the calculations shown in figure S10, it would be good to know what the EQE curve looks like for longer length (given that the tradeoff here is likely to be between devices of longer length and larger W_{gap} and those with close W_{gap} and shorter length). The tradeoff with length and speed (if meaningful) would then be nice to know to allow for full device design.

Reply: Thanks for the comment. This comment now refers to Fig. S12. We have added the EQE simulation for longer device lengths (30 and 50 μm) in Fig. S12(b), as shown below (Fig. R5). Similar to the 10- μm long device, the EQE first increases with W_{gap} due to the reduced perturbation when entering the Si, and then decreases due to the reduced leaky-wave coupling efficiency.

Unfortunately, a comprehensive study between the device length and speed was limited by the instrumentation as seen in Fig. S15. We have also measured the impulse response of 500- μm long devices (PIN) and found the 3-dB bandwidth was still limited by the instrumentation. Shorter pulse-width lasers or high-speed visible spectrum modulators can be used for more comprehensive bandwidth characterization.

A general device design rule can be to first determine the maximum length needed for the desired OE bandwidth and then choose the W_{gap} (if possible) to maximize the EQE. Our results show that $W_{\text{gap}} = 190 \text{ nm}$ approached EQE saturation at a device length of 50 μm .

We have also included the above discussion into the Supplementary Information (Section S8, highlighted in yellow) as shown below:

Figure S12 shows the effect of W_{gap} , the interlayer spacing, on the mode mismatch loss and EQE. Due to the loss of mode confinement in SiN in the presence of Si, $\eta_{\text{mode}} \rightarrow 0$ as $W_{\text{gap}} \rightarrow 0$. $\eta_{\text{mode}} \rightarrow 1$ as the Si is separated from the SiN. The EQE peaks at W_{gap} of $\sim 110\text{-}220 \text{ nm}$ for both TE and TM polarizations. The EQE decrease is due to the mode mismatch at small W_{gap} and lower evanescent coupling at large W_{gap} . To achieve both a high EQE and speed (i.e., at a short device length), a general rule can be to first determine the maximum length that can be used, and then choose a W_{gap} to maximize the EQE.

8. There are a number of small typos and the like that should be fixed:

a. In figure 1A the “SiN 1” label is somewhat confusingly placed and should be moved to be closer to the bottom layer

Reply: Thanks for the comment. Fig. 1 has been revised.

b. Figure S4 should be redone to make clearer which line corresponds to which length and that two of the blue lines are on top of each other

Reply: Thanks for the comment. This figure has been amended.

c. Loc 54 in the supplemental material: ?? -> 5

Reply: Thanks for the comment. We have corrected this in the revised manuscript.

d. The dashed curves are not visible in figure S7. Also, it would be nice if the maximum possible responsivity was noted in the figures.

Reply: Thanks for the comment. We have amended Fig. S9 and indicated the maximum responsivity (i.e., 100% EQE) at each wavelength.

e. This is a bit nit-picky, but it would be nice to see the whole forward bias curve in Fig S13 and not just the linear part.

Reply: Thanks for the comment. We have amended Fig. S15 to show the whole forward bias range.

f. It would be helpful to include the APD OE bandwidth in the section of the main paper that starts on line 201. This prevents the reader from having to hunt for it.

Reply: Thanks for the comment. We have included the APD OE bandwidth as below in the revised manuscript:

We then found an avalanche multiplication gain, M , of 46 ± 14 , OE 3-dB bandwidth of 4.0 ± 1.4 GHz and a corresponding gain-bandwidth product (GBP) of 173 ± 30 GHz at the avalanche breakdown voltage, V_{br} (Fig. 2(g)).

g. The refractive indices are given, but it is not specified how they were measured. This should be added (Loc 285).

Reply: Thanks for the comment. The refractive indices were measured using ellipsometry. We have amended the main text as below (Loc...):

Ellipsometry measurements show SiN and cladding SiO₂ had refractive indices of ~ 1.82 and ~ 1.46 , respectively, at $\lambda = 488$ nm.

The following things would make the paper stronger, but (in my opinion) do NOT need to be addressed to allow for publication:

1. The measurement and analysis of the high-speed and APD performance of the device was less thorough than the rest of the paper. This is to be expected as these measurements are more complex and often limited by available equipment.

Reply: Thanks for the comment. The high-speed measurement of the devices is limited by our lack of a high-speed EO modulator and a high-bandwidth oscilloscope. If these can be improved, the OE bandwidth of the PDs can be studied in more detail.

a. It would be nice if the speed could be characterized in both polarizations or for more than just 405 nm light. If this is not possible, some discussion on whether the speed would be expected to vary with polarization (presumably not) or wavelength (mildly?) would add to the paper.

Reply: Thanks for the comment. We have discussed in Question 2 that the polarization should not influence the OE bandwidth, since the OE bandwidth is determined by electrical parameters such as the electric field in the PN/PIN junction, device series resistance and total capacitance. Since the 405-nm pulsed laser had the shortest pulse width we could obtain at the time of measurement, and the OE

bandwidth was later verified to be limited by the pulse width and the oscilloscope, we decided not to characterize the device at other wavelengths with longer pulse widths.

We expect the OE bandwidth to be compromised at longer wavelengths (at least at low bias). This can be due to the longer carrier transit time by the diffusion of photon-generated carriers deeper below the junction (lower Si absorption coefficient at longer wavelengths), provided that the bandwidth is limited by the carrier transit time at this bias range.

b. Similarly, the paper would be strengthened if both the linearity and speed data had been taken at the same wavelength to provide a complete characterization at one wavelength. I assume that this wasn't done due to instrument limitations.

Reply: Thanks for the comment. Yes, having the linearity measurement at 405 nm would complete all PD characterizations at one wavelength at 405 nm. We also have the linearity data collected for one of the 50- μm PIN devices at 405 nm, as shown below, but with fewer data points than 488 nm (so the plot is not included in the manuscript). The device exhibited similar linearity as that at 488 nm (Fig. 2(c)).

Fig. R1 I_{eph} as a function of $P_{in,PD}$ at $\lambda = 405$ nm.

c. Given that the material parameters of silicon are very well understood, it should be straight forward to simulate the expected carrier transit times in, say, Sentaurus for this device and determine if they match the measured speeds at low bias (before the measurement system becomes the limiting factor or avalanching starts (where simulation gets more tricky)). I'm not sure why this wasn't done.

Reply: Thanks for the comment. This is a good suggestion and can be included in future work.

d. The APD characterization could include the excess noise figure, but is also good as is.

Reply: Thanks for the comment. We will include this study in future work.

2. I agree with the authors that a dopant well structure can be used to help suppress the device cross-talk, but think the picture they have presented in section S12 is a overly simplistic, and I'm not sure it's adding anything. I would consider either cutting section S12 and leaving the comment at loc 265 to stand on its own, or expanding it (though I don't think either is necessary for publication).

Reply: Thanks for the comment. We have removed this section.

On the whole, I liked this paper and support its publication in Nature Communications.

References

[R1] S. Yanikgonul, V. Leong, J. R. Ong, T. Hu, S. Y. Siew, C. E. Png, and L. Krivitsky, *Nature communications* 12, 1834 (2021).

[R2] S. Cuyvers, A. Hermans, M. Kiewiet, J. Goyvaerts, G. Roelkens, K. Van Gasse, D. Van Thourhout, and B. Kuyken, *Opt. Lett.* 47, 937-940 (2022).

[R3] R. Morgan, D. Kharas, J. Knecht, P. Juodawlkis, K. Cahoy, and C. Sorace-Agaskar, *Waveguide-integrated blue light detector*, in *2021 IEEE Photonics Conference (IPC)* (2021) pp. 1-2.

[R4] Y. Guangwei, R. Pownall, P. Nikkel, C. Thangaraj, T. W. Chen, and K. L. Lear, *IEEE Photonics Technology Letters* 18, 1657-1659 (2006).

[R5] C. De Vita, F. Toso, N. G. Pruiti, C. Klitis, G. Ferrari, M. Sorel, A. Melloni, and F. Morichetti, *arXiv preprint arXiv:2202.04413* (2022).

[R6] R. Gherabli, S. Indukuri, R. Zektzer, C. Frydendahl, and U. Levy, *arXiv preprint arXiv:2112.08920* (2021).

[R7] M. Jafari, T. Fatema, D. R. Carlson, S. B. Papp, and A. Beling, *Heterogeneous integration of algaas/gaas photodiodes on tantalum waveguides for visible-light applications*, in *Conference on Lasers and Electro-Optics* (Optica Publishing Group, 2022) p. STu5G.5.

REVIEWERS' COMMENTS

Reviewer #2 (Remarks to the Author):

The authors have sufficiently addressed my comments, and I believe that this paper should be published. Congratulations to the authors on a great paper.